# Associations of Diabetes and Hyperglycaemia with Extent and Outcomes of Acute Burn Injuries

**DOI:** 10.3390/biomedicines12051127

**Published:** 2024-05-19

**Authors:** Jeffrey Chandra, Edward Raby, Fiona M. Wood, P. Gerry Fegan, Bu B. Yeap

**Affiliations:** 1Medical School, University of Western Australia, Perth 6009, Australia; 2State Adult Burns Unit, Fiona Stanley Hospital, Perth 6150, Australia; 3Department of Endocrinology and Diabetes, Fiona Stanley Hospital, Perth 6150, Australia; 4Medical School, Curtin University, Perth 6102, Australia

**Keywords:** diabetes, stress hyperglycaemia, burns, hospitalisation

## Abstract

Background: Severe burns may induce hyperglycaemia in the absence of diabetes, but how glucose trajectories relate to burns outcomes is unclear. Aim: To assess incidence of hyperglycaemia following acute burn injury, and associations with diabetes history and length of stay (LOS). Methods: Retrospective cohort study of adults admitted with acute burns to tertiary centres. Blood glucose level (BGL), hyperglycaemic episodes (BGL ≥ 11.1 mmol/L) and hyperglycaemic days were recorded. Stress hyperglycaemia was defined as BGL ≥ 11.1 mmol/L without a diabetes history. Results: A total of 30 participants had a diabetes history and 260 did not. Participants with known diabetes had higher mean BGLs (9.7 vs. 9.0 mmol/L, *p* < 0.001), more hyperglycaemic episodes (28.0 vs. 17.2%, *p* < 0.001) and hyperglycaemic days (51 vs. 21%, *p* < 0.001), compared to those without diabetes, despite smaller burns (total body surface area 1.0 vs. 14.8%, *p* < 0.001). Fourteen participants with stress hyperglycaemia had similar BGLs (at admission 10.3 vs. 11.5 mmol/L; during inpatient stay 9.9 vs. 9.8 mmol/L), more severe burns (15.6% vs. 1.0% TBSA) and longer LOS (18 vs. 7 days, *p* < 0.001) compared to participants with known diabetes. Extent of burns, having NGT nutrition, age, having inpatient BGL monitoring in the setting of diabetes, or having inpatient BGL monitoring in the absence of diabetes were associated with longer LOS. Conclusions: In participants with known diabetes, small burn injuries were associated with hyperglycaemia. Stress hyperglycaemia can be triggered by major burn injuries, with early and sustained elevation of BGLs. Further research is warranted to improve inpatient management of BGL in patients with acute burn injury.

## 1. Introduction

Burn injuries are a major cause of trauma with larger injuries often requiring prolonged admissions in specialised burn centres for intensive wound care and rehabilitation [1]. Long-term sequelae include poorer quality of life, depression and pain, and lost work productivity [2,3,4]. Severe burns trigger a systemic inflammatory response with release of stress hormones inducing a hypermetabolic state [5,6]. Release of glucocorticoids promotes gluconeogenesis, while catecholamines stimulate proteolysis, lipolysis and hepatic gluconeogenesis and glycogenolysis [7,8]. These mechanisms increase glucose production, reduce clearance, and along with impaired insulin signalling reducing cellular glucose uptake, predispose to sustained hyperglycaemia [9].

Stress hyperglycaemia can be defined as fasting glucose > 6.9 mmol/L or random glucose > 11.1 mmol/L in an acutely ill patient without evidence of previous diabetes, or as a deterioration of pre-existing diabetes [10]. Literature describing stress hyperglycaemia in the context of burn injuries is sparse. In a paediatric burns cohort, non-fasted serum glucose was significantly elevated for up to 60 days post injury, without exceeding the threshold of >11.1 mmol/L [11]. In an adult intensive care unit burns population, up to 17% of days included a random glucose level >11.1 mmol/L; however, pre-existing diabetes status was not reported [12]. Thus, further investigation of the extent and drivers of dysglycaemia (hyperglycaemia, hypoglycaemia and glucose variability) and their relationship to outcomes following acute burn injuries is warranted [13].

Acute hyperglycaemia in critical illness, including in participants without a history of diabetes, is associated with poorer outcomes, with one study showing that maintaining blood glucose within 4.4–6.1 mmol/L was associated with lower mortality compared to tolerating levels of 11.9 mmol/L [14]. However, subsequent studies comparing intensive to moderate glucose control demonstrated increased hypoglycaemia and possibly increased mortality [15,16]. Chronic hyperglycaemia, in the setting of diabetes, interferes with healing. Mechanisms include alteration of immune cell function leading to greater infection risk and inflammation of autonomic neurons precipitating microvascular dysfunction and tissue hypoperfusion [17,18,19]. Previous studies comparing adult burn participants with and without diabetes have documented added complications in the setting of diabetes, noting longer admissions and overall morbidity, including wound infections, sepsis and organ failure [17,18]. However, guidelines for recommended glycaemic targets vary: some authors suggest maintaining blood glucose levels at 7.2–8.3 mmol/L, although to assist practical implementation others suggest 5–10 mmol/L [20,21].

We examined the extent of dysglycaemia in burns inpatients with known diabetes and of stress hyperglycaemia in participants without a history of diabetes, and analysed factors associated with hyperglycaemic events in this setting.

## 2. Materials and Methods

### 2.1. Derivation of the Study Cohort

Electronic medical records were retrieved for all adults admitted to the Fiona Stanley Hospital (FSH) Burns Unit with a principal diagnosis of acute burn injury over the period July 2020–June 2021 inclusive. Participants admitted for follow-up surgical management (repeat debridement or reconstruction) or with different principal diagnosis on their discharge summary were excluded. This study was approved as a Quality Improvement Project by the SMHS Governance committee (GEKO number 41936).

### 2.2. Definition of Diabetes

Participants were categorised by historic documentation of diabetes, as either “Known Diabetes” or “No History of Diabetes”. In supplementary analyses involving only participants with HbA1c on admission, those with no history of diabetes but admission HbA1c ≥ 6.5% were categorised as having undiagnosed diabetes.

### 2.3. Variables of Interest

Date of admission, sex, age, weight, percentage of total body surface area affected by burns (%TBSA), site of burns injury, nutrition route, length of stay (LOS) and medications on admission were retrieved from electronic medical records. Any history of type 1 or 2 diabetes was noted. Pre-admission glucose-lowering medications, glucose-lowering medications prescribed during admission, HbA1c and blood glucose level (BGL) on admission, and BGL measurements during inpatient stay (from admission until discharge, or for the first 14 days of longer admissions) were recorded. Other variables of interest were the proportion of hyperglycaemic and hypoglycaemic episodes (BGL ≥ 11.1, ≤4 mmol/L) [22], median days with BGL monitoring, proportion of hyperglycaemic days (patient days with BGL ≥ 11.1 mmol/L), and LOS [18].

### 2.4. Statistical Analysis

The primary analysis was conducted based on diabetes status from history only (known diabetes vs. no history of diabetes, 30 vs. 260). A secondary analysis was conducted for diabetic status based on history or admission HbA1c (known diabetes vs. undiagnosed diabetes vs. no diabetes, 30 vs. 4 vs. 117). Data are shown as % (N), with median (inter-quartile range) used for cohort demographics and mean (SD) for BGL data. Comparisons between age, %TBSA, nutrition route (oral vs. NGT), LOS and admission HbA1c were conducted using the Mann–Whitney U test. Proportions were compared through two proportion test and 4, 5 and 6-sample test for equality of proportions. Length of stay (LOS) was modelled as count data. Both Poisson and quasi-Poisson regression were determined to be suboptimal for modelling LOS as an outcome due to overdispersion (dispersion parameter 2.91). Hence, negative binomial regression modelling was performed.. Participants with missing data in any predictor were excluded from that analysis. Initial univariate regression models were constructed; odds ratio coefficients, 95% confidence intervals and *p*-values were calculated. Multivariate models were constructed through backward selection, ceasing when the Akaike Information Criterion (AIC) value was not improved with elimination of further variables. Two-sided *p* values < 0.05 were considered significant. Analyses were completed on RStudio, Version 2022.2.1.461 [23].

## 3. Results

### 3.1. Baseline Characteristics and BGL Monitoring

Of 30 participants with known diabetes, 29 had BGL monitoring during inpatient stay (Figure 1). Of 260 participants with no history of diabetes, 34 received BGL monitoring during inpatient stay, with varied rationales for monitoring. All 22 participants with TBSA between 11.9% and 52% received BGL monitoring. Twelve participants with TBSA < 11.9%, were monitored, of whom three had an admission HbA1c ≥ 6.5% (undiagnosed diabetes), and two whilst in the intensive care unit (ICU). Of the remaining seven, two were obese (BMI 34.2 and 38.2 kg/m^2^) and two were monitored following alcohol withdrawal protocols.

### 3.2. Comparison of Participants with Known Diabetes vs. No History of Diabetes

Participants with a history of diabetes were older (age 55 vs. 49 years, *p* < 0.01), had smaller burns (TBSA 1.0 vs. 14.8%, *p* < 0.001), smaller proportions of NGT nutrition (3.3 vs. 62%, *p* < 0.001), shorter LOS (6 vs. 17 days, *p* < 0.001) and higher admission HbA1c (8.4 vs. 5.6%, *p* < 0.001) compared to those without a history of diabetes who had BGL monitoring (Table 1).

Similar differences were found comparing those with known diabetes to those without a history of diabetes who did not have BGL monitoring. Those with known diabetes were older (55 vs. 38 years, *p* < 0.001), had smaller burns (%TBSA 1.0 vs. 2.1%, *p* = 0.01) and higher admission HbA1c (8.4 vs. 5.2%, *p* < 0.001) but had longer LOS (6 vs. 5 days, *p* = 0.03) compared to those with no history of diabetes who were not monitored (Table 1).

### 3.3. Hyperglycaemic Episodes and Hyperglycaemic Days

Participants with known diabetes had higher mean BGLs during hospitalisation (9.7 vs. 9.0 mmol/L, *p* < 0.001) with greater proportions of hyperglycaemic episodes (BGL ≥ 11.1 mmol/L, 28.0 vs. 17.2%, *p* < 0.001) and hyperglycaemic days (days with BGL ≥ 11.1 mmol/L, 51 vs. 21%, *p* < 0.001), compared to those without a history of diabetes who had BGL monitoring. However, the cohorts had similar median number of days with BGL monitoring (6 vs. 8, *p* = 0.65) (Table 1). 

For participants without a history of diabetes, those who had their BGLs monitored had larger burns (TBSA 14.8 vs. 2.1%, *p* < 0.001), longer LOS (17 vs. 5 days, *p* < 0.001) and higher admission HbA1c (5.6 vs. 5.2%, *p* < 0.01) than those who were not.

### 3.4. Stress Hyperglycaemia in Participants without Known Diabetes

Approximately 40% (14/34) of participants with no history of diabetes who received BGL monitoring developed at least one episode of stress hyperglycaemia (BGL ≥ 11.1 mmol/L). These participants had larger burn injuries (TBSA 15.6 vs. 1.0%, *p* < 0.001), lower admission HbA1c (5.7 vs. 8.6%, *p* < 0.001) and lower proportion of NGT nutrition (4.2 vs. 79%, *p* < 0.001) but similar admission BGLs (10.3 vs. 11.5 mmol/L, *p* = 0.13), compared to participants with known diabetes who experienced hyperglycaemia (Table 2). Participants experiencing stress hyperglycaemia also had higher mean BGLs (9.92 vs. 9.86 mmol/L, *p* = 0.02) and longer LOS (18 vs. 7 days, *p* < 0.001) compared to those with known diabetes.

In participants experiencing hyperglycaemia, there was no difference in the proportion of hyperglycaemic events (BGL ≥ 11.1 mmol/L: 29.8 vs. 27.9%, *p* = 0.52; BGL ≥ 16 mmol/L: 7.1 vs. 8.2%, *p* = 0.52) or the median number of days with BGL monitoring (6.5 vs. 10 days, *p* = 0.09) between participants with and without known diabetes. However, those with known diabetes experienced more hypoglycaemic events ≤4 mmol/L (2.8 vs. 0.2%, *p* = 0.002) and hyperglycaemic days (56.0 vs. 38.7%, *p* < 0.01). Hypoglycaemic episodes occurred exclusively in participants on insulin therapy, including three participants with T1DM.

Participants without diabetes experiencing stress hyperglycaemia (BGL ≥ 11.1 mmol/L) had similar burn severity (TBSA 15.6 vs. 14.2%, *p* = 0.58), LOS (18 vs. 17 days, *p* = 0.58) and admission HbA1c (5.7 vs. 5.4%, *p* = 0.13) compared to those who did not (Table 2). They were older and had greater proportion of NGT nutrition, but the difference was not statistically significant (50 vs. 41 years, *p* = 0.60, 79 vs. 50%, *p* = 0.184). Those with stress hyperglycaemia had higher admission BGLs (10.3 vs. 7.4 mmol/L, *p* = 0.01), higher BGLs throughout inpatient stay (mean BGL 9.9 vs. 7.6 mmol/L, *p* < 0.001) and BGLs monitored for longer (Median 10 vs. 4.5 days, *p* = 0.005). Eleven of twelve recorded admission BGL ≥ 7.8 mmol/L, including five who required insulin to control their blood glucose while one required metformin monotherapy (for further details on glucose-lowering medications see Appendix A).

### 3.5. Burn Severity and Occurrence of Dysglycaemia in Participants with Known Diabetes

Of participants with known diabetes, half had burn injuries <1% TBSA (Table 3A). Over a quarter of inpatient BGLs were ≥11.1 mmol/L. The proportion of hyperglycaemic episodes was comparable across all TBSA categories (*p* = 0.17) (Table 3A). Over half of BGL monitored days included a BGL ≥ 11.1 mmol/L. This proportion was similar across TBSA categories (*p* = 0.58). Of 12 participants requiring insulin during their stay, 11 participants had TBSA < 2%, including 2 participants who were not on insulin prior to admission.

### 3.6. Burn Severity and Occurrence of Hyperglycaemia in Participants without Known Diabetes

A sixth of inpatient BGLs were ≥11.1 mmol/L, with significance difference in the proportion of episodes across TBSA intervals (<1, 2–5, 5–10 and ≥10%: 0 vs. 15 vs. 53 vs. 11%, *p* < 0.001) (Table 3B). Larger burn injuries were associated with more days with BGL ≥11.1 mmol/L (TBSA 2–5, 5–10 and ≥10%: 23 vs. 61 vs. 16%, *p* < 0.001). The higher proportion of hyperglycaemic episodes and days with BGL ≥11.1 mmol/L in participants with TBSA 5–10% was attributable to two participants with undiagnosed diabetes (admission HbA1c 6.7 and 12.9%).

### 3.7. Factors Associated with LOS

In multivariate negative binomial analysis, extent of burns, having NGT nutrition, age, having diabetes, or having inpatient BGL monitoring in the absence of diabetes were all significant predictors of longer length of stay. (Table 4). Larger burn injuries were associated with incrementally longer hospital admissions (1.02 days per % TBSA, 95% CI 1.01, 1.04, *p* < 0.001). Prompted BGL monitoring predicted extended length of stay in those with known diabetes (1.44 days, *p* < 0.003), and in those receiving BGL monitoring without a history of diabetes (1.53 days, *p* < 0.008).

### 3.8. Supplementary Analyses

Through admission HbA1c ≥ 6.5%, an additional four participants with previously undiagnosed diabetes were identified (Appendix A). Results from analyses of factors associated with hyperglycaemia and hyperglycaemic days between participants with and without diabetes remained largely unchanged (Appendix A). Once stratified by HbA1c status, variables modelling LOS remained largely the same (Appendix A). Participants with previously undiagnosed diabetes experienced difficult BGL control with 59% of events being ≥11.1 mmol/L and hyperglycaemia persisting through 80% of patient days (Appendix A).

In participants with no diabetes, confirmed by HbA1c <6.5%, 14% of measured BGL events were ≥11.1 mmol/L, with 20% of days with BGL monitoring having at least one stress hyperglycaemic event (Appendix A). Eight of eighteen participants without diabetes who had BGL monitoring experienced stress hyperglycaemia (Appendix A). Stress hyperglycaemia occurred mostly in participants with TBSA ≥ 10% (Appendix A).

## 4. Discussion

In participants with known diabetes admitted acutely with relatively small burn injuries, hyperglycaemic events were common and occurred on half of inpatient days. In 14 of 34 participants with no history of diabetes who had BGL monitoring, more severe burn injuries precipitated stress hyperglycaemia, resulting in elevated BGLs comparable to levels in participants with known diabetes. The presence of known diabetes was associated with more hyperglycaemic days. Burn severity, age, having NGT nutrition and prompted inpatient BGL monitoring by way of known diabetes or clinical concern in those without diabetes were associated with longer hospital admissions.

Previous studies have identified existing diabetes as a factor driving infection and sepsis in burn injuries [17,18]. The mechanisms contributing to poorer outcomes are multifactorial [7]. Hyperglycaemia may suppress anti-inflammatory cytokines, impairing neutrophil and macrophage action [24]. Chronic hyperglycaemia predisposes to vascular disease, limits neovascularisation and wound healing, and impairs skin and anastomotic healing [25,26,27]. While these potential adverse effects of hyperglycaemia in the setting of burns are recognised, information on glucose concentrations during hospitalisations for acute burn injuries is limited. Studies have investigated inpatient glucose trajectories in acute burns; however, papers comparing trends according to diabetic status are lacking [11,12,28]. In our study, in participants with known diabetes, 28% of BGLs measured were ≥11.1 mmol/L and 51% of in-hospital days had at least one elevated BGL, representing a substantive burden of hyperglycaemia. This is despite 19/29 participants with diabetes having TBSA of <2% with similar rates of hyperglycaemic events and hyperglycaemic days across the range of TBSA categories.

Tools such the Baux score demonstrate that burn injuries follow a characteristic dose-related response where larger injuries lead to poorer outcomes [29]. However, for participants with known diabetes, our data indicate stimuli for hyperglycaemia following acute burn injuries are not related to burn severity, as determined using TBSA.

There are limited descriptions of burn-induced stress hyperglycaemia in the current literature. Previous studies have demonstrated elevated blood glucose in burns populations however these have not addressed the definitions of stress hyperglycaemia proposed by Dungan [10,11,12]. We found that hyperglycaemia (BGL > 11.1 mmol/L) occurred in adult burn participants with no history of diabetes, albeit in settings of larger burns. The degree of hyperglycaemia was comparable to participants with known diabetes, with similar mean BGL and frequency of hyperglycaemic events. Factors predicting burn-related stress hyperglycaemia have not been well described, impeding the development of inpatient glucose monitoring plans. A key finding of the current study is that stress hyperglycaemia may be detectable early during inpatient stay, as admission BGLs were already elevated. 

Participants who had stress hyperglycaemia had a mean admission BGL of 10.3 mmol/L, almost invariably ≥7.8 mmol/L. While further characterisation of the glucose trends in burns participants with no history of diabetes is needed, admission BGLs could be incorporated into an admission/inpatient BGL monitoring plan to detect stress hyperglycaemia.

Previous studies have demonstrated the importance of normalising glucose in critical illness, including burns [14,20]. In participants with diabetes, use of insulin was predominantly in small %TBSA. For participants without diabetes, insulin was not required for injuries <12%TBSA. Importantly, those with known diabetes experienced more difficult glucose control, with hyperglycaemic episodes across more patient days. 

Based on our observations, participants with known diabetes warrant BGL monitoring and optimisation of diabetes therapy even with relatively small burn injuries. Our results suggest that in the absence of a diabetes history, admission HbA1c may identify a small but clinically important number of participants who would benefit from intensive BGL monitoring and management. Thus, routine measurement of HbA1c to detect undiagnosed diabetes should be considered in acute burns.

In addition to greater extent of burn injury, age, and NGT nutrition, monitoring BGL in patients with diabetes and in those without diabetes were associated with longer LOS, The associations of BGL monitoring with LOS may reflect more complex admissions and higher levels of clinical concern. While BGL monitoring can be readily justified in patients with diabetes, indicators for BGL monitoring in participants without known diabetes might be larger burn injury (TBSA ≥ 5%) and admission BGL (≥7.8 mmol/L), for detection of stress hyperglycaemia. An admission (random) BGL of ≥7.8 mmol/L identified almost all our participants who did not have a history of diabetes, who developed stress hyperglycaemia. More severe burn injuries (TBSA ≥ 12%) were associated with difficult glucose control, increasing the likelihood of requiring intensive glucose-lowering treatment. While a previous study suggested that metformin alone could control glucose concentrations as effectively as insulin in the burns setting [30], in our institution insulin was not infrequently employed to counteract stress hyperglycaemia. Further investigation would be required to ascertain the optimal threshold for instituting BGL monitoring and the optimal duration of such monitoring on burns participants without a history of diabetes.

Strengths of our study are the analysis of a substantial cohort of burns participants from a tertiary centre, with burns across a range of severity [31], as well as the characterisation of diabetes status in conjunction with analysis of BGLs during inpatient stays. Limitations include its cross-sectional nature, and limited numbers of participants experiencing stress hyperglycaemia. Only a subset of participants had HbA1c measured, leading to unknown pre-injury glucose control. We used retrospectively collected BGL data and in our institution, continuous glucose monitoring was not used. Additionally, there may be variability in BGL monitoring practice between burns and endocrinology departments. Larger studies are needed to confirm our findings, with extended follow-up to clarify the impact of burn injury on longer-term glycaemic control.

In conclusion, in participants with known diabetes, relatively small burn injuries may result in hyperglycaemia. Stress hyperglycaemia can occur as a result of substantive burn injuries, resulting in promptly elevated BGLs. There is scope to improve inpatient management of BGLs and to investigate whether this would improve longer-term clinical outcomes following burns.

## Figures and Tables

**Figure 1 biomedicines-12-01127-f001:**
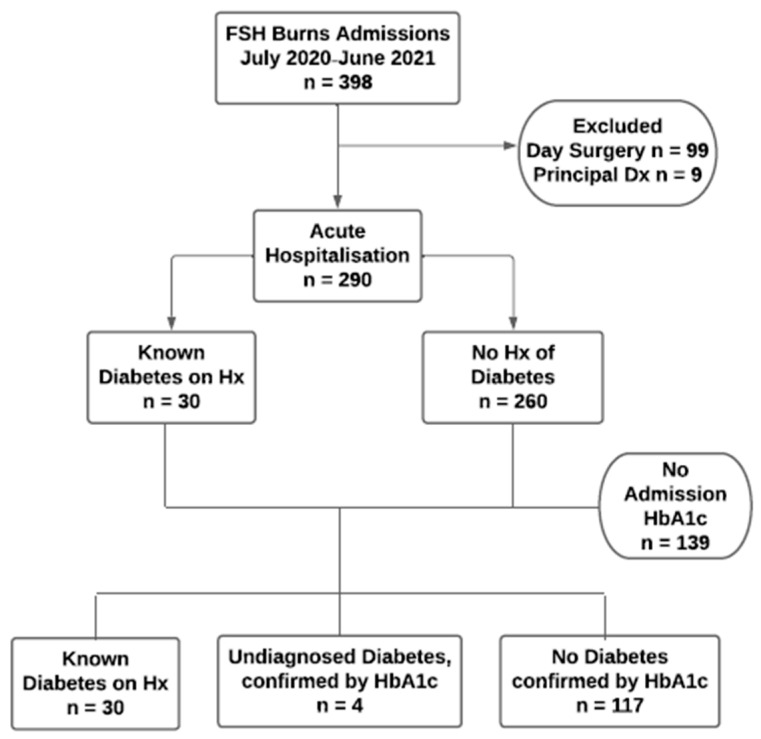
Flowchart showing derivation of the study population.

**Table 1 biomedicines-12-01127-t001:** Baseline characteristics. Data are shown as median (25th and 75th percentiles), mean (SD) or percentage (N).

	Known Diabetes *	No History of Diabetes	
		BGL Monitoring	No BGL Monitoring
N	30 (M = 22, F = 8)	34 (M = 24, F = 10)	226 (M = 154, F = 72)
Age (years)	55 (47, 71)	49 (35, 57)	38 (26, 53)
%TBSA (%)	1.0 (0.5, 3.0)	14.8 (7.1, 23.5)	2.11 (1.0, 4.4)
% patients on NGT nutrition	3.3% (1)	62% (21)	0% (0)
LOS (days)	6 (4.0, 10.0)	17 (10.3, 32.8)	5 (3.0, 7.0)
Admission HbA1c (%)	8.4 (6.9, 9.5)	5.6 (5.3, 5.9)	5.2 (5.0, 5.6)
Total BGL measurements	679	752	0
Mean BGL (SD)	9.7 (3.7)	9.0 (3.5)	N/A
% patients with any BGL > 11.1 mmol/L (N)	80.0% (24)	41.2% (14)	N/A
% BGL ≥ 11.1 mmol/L (N)	28.0% (190)	17.2% (129)	N/A
% patients with any BGL > 16 mmol/L (N)	43.3% (13)	14.7% (5)	N/A
% BGL ≥ 16 mmol/L (N)	6.6% (45)	5.1% (38)	N/A
% patients with any BGL < 4 mmol/L (N)	16.7% (5)	2.9% (1)	N/A
% BGL ≤ 4 mmol/L (N)	2.7% (18)	0.1% (1)	N/A
Median days with BGL monitoring	6 (3.8, 10)	8 (3, 11.2)	N/A
Total days with BGL measurement	201	248	0
% of days with BGL ≥ 11.1 mmol/L	51% (102)	21% (53)	N/A

* 29 patients with known diabetes had BGL monitoring. BGL: Blood glucose. TBSA: Total body surface area. NGT: Nasogastric tube. LOS: Length of stay.

**Table 2 biomedicines-12-01127-t002:** Characteristics of patients who had BGL monitoring. Data are shown as median (25th and 75th percentiles), mean (SD) or percentage (N).

	Known Diabetes	No History of Diabetes	No History of Diabetes, Did Not Experience Hyperglycaemia
	Experienced Hyperglycaemia (BGL ≥ 11.1 mmol/L)		
N	24 (M = 18, F = 6)	14 (M = 10, F = 4)	20 (M = 14, F = 6)
Age (years)	53 (44, 67)	50 (46, 54)	41 (27, 59)
%TBSA (%)	1.0 (0.5, 2.5)	15.6 (10.1, 23.5)	14.2 (5.0, 22.8)
% patients on NGT nutrition	4.2% (1)	78.6% (11)	50.0% (10)
LOS (days)	7 (4.7, 11.0)	18 (12.5, 32.8)	17 (9.2, 26.2)
Admission HbA1c (%)	8.6 (7.9, 9.5)	5.7 (5.4, 6.5)	5.4 (5.2, 5.8)
Total inpatient BGL measurements	22	12	20
Mean inpatient BGL (SD)	11.5 (3.46)	10.3 (3.7)	7.4 (1.5)
Total BGL Measurements	637	463	289
Mean BGL (SD)	9.8 (3.7)	9.9 (4.09)	7.6 (1.41)
% BGL ≥ 11.1 mmol/L (N)	29.8% (190)	27.9% (129)	N/A
% BGL ≥ 16 mmol/L (N)	7.1% (45)	8.2% (38)	N/A
% BGL ≤ 4 mmol/L (N)	2.8% (18)	0.2% (1)	0% (0)
Median days with BGL monitoring	6.5 (4.2, 10)	10 (5.7, 14)	4.5 (2.0, 8.0)
Total days with BGL measurement	182	137	111
% of days with BGL ≥ 11.1 mmol/L	56.0% (102)	38.7% (53)	0% (0)

**Table 3 biomedicines-12-01127-t003:** Patients with known diabetes (A) and without a history of diabetes (B) who received inpatient BGL monitoring, stratified according to %TBSA. Data are shown as percentage (%),number (n) of BGL events and days with BGL ≥ 11.1 mmol/L.

(A)							
%TBSA	N	Total BGLs	% of BGL Events (n)			Total Days with BGLs	Days with BGL ≥ 11.1 mmol/L
			≥11.1 mmol/L	≥16 mmol/L	≤4 mmol/L		
<1	14	307	31% (95)	6.8% (21)	2.0% (6)	85	54% (46)
1 to <2	5	150	27% (40)	8.7% (13)	0% (0)	47	45% (21)
2 to <5	4	83	19% (16)	4.8% (4)	2.4% (2)	26	42% (11)
5 to <10	2	85	24% (20)	3.5% (3)	24% (20)	24	63% (15)
≥10	3	46	33% (15)	8.7% (4)	2.2% (1)	16	44% (7)
Unknown	1	8	50% (4)	0% (0)	0% (0)	3	67% (2)
Total	29	679	28% (190)	6.6% (45)	4.3% (29)	201	51% (102)
**(B)**							
<1	2	11	0% (0)	0% (0)	0% (0)	5	0% (0)
1 to <2	0	0	0% (0)	0% (0)	0% (0)	0	0% (0)
2 to <5	5	33	15% (5)	3.0% (1)	3.0% (1)	13	23% (3)
5 to <10	5	113	53% (60)	21% (24)	22% (25)	31	61% (19)
≥10	22	595	11% (64)	2.2% (13)	2.2% (13)	199	16% (31)
Unknown	0	0	0	0	0	0	0
Total	34	752	17% (129)	5.1% (38)	5.2% (39)	248	21% (53)

**Table 4 biomedicines-12-01127-t004:** Associations of variables with risk of length of stay (LOS) during hospitalisation for acute burn injury (N = 283). Data are shown as regression coefficients for change in outcome counts using Negative Binomial regression, with 95% confidence intervals, and *p*-values for individual factors.

Variables *	Negative Binomial ^a^
**Intercept**	3.92 (3.22, 4.77)
**Diabetes/monitoring Status**	
No history of diabetes, BGL not monitored	Reference
No history of diabetes, BGL monitored	1.53 (1.14, 2.05), *p* < 0.008
Known diabetes, BGL monitored	1.44 (1.13, 1.83), *p* < 0.003
**Inpatient hyperglycaemia**	
Inpatient hyperglycaemia	
**Nutrition Route**	
Oral	Reference
NGT	2.39 (1.65, 3.46), *p* < 0.001
**Burn Injury (TBSA)**	
TBSA	1.02 (1.01, 1.04), *p* < 0.001
**Age**	
Age	(1.00, 1.01), *p* < 0.004
**Sex**	
Male	
Female	
**AIC**	1548.8

* Backward selection was performed to create further models. Selection ceased when AIC was not improved with further elimination. ^a^ Multivariate negative binomial regression model.

## Data Availability

Data is contained within the article and Appendix A.

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
