# Peer review of "Associations of Diabetes and Hyperglycaemia with Extent and Outcomes of Acute Burn Injuries"

_biomedicines, 2024, doi:10.3390/biomedicines12051127_

Round 1
Reviewer 1 Report
Comments and Suggestions for Authors
This is an interesting manuscript to elucidate the associations of diabetes and hyperglycemia with length of stay of acute burn patients. The manuscript can be considered for publication after addressing the following comments.
Comments:
1. In the abstract, what is the different between admission and during admission in the statement "(admission 10.3 vs 11.5 mmol/L; during admission 9.9 vs 9.8 20 mmol/L)"?
2. Use either BGL or BSL, not both, consistently in the manucript.
3. In the table 1 or in a supplementary table, the authors should indicate the number and percentage of patients in each group with BSC >11.1, >16, and <4.
4. In table 2, are there any patients with known diabetes but did not experienced hyperglycemia?
Author Response
1. Summary |
|
|
Thank you very much for taking the time to review this manuscript. Please find the detailed responses below and the corresponding revisions in tracked changes in the re-submitted files, alongside a finalized manuscript incorporating all reviewer recommendations.
|
||
2. Questions for General Evaluation |
Reviewer’s Evaluation |
Response and Revisions |
Does the introduction provide sufficient background and include all relevant references? |
Yes |
|
Are all the cited references relevant to the research? |
Yes |
|
Is the research design appropriate? |
Can be improved |
|
Are the methods adequately described? |
Yes |
|
Are the results clearly presented? |
Can be improved
|
|
Are the conclusions supported by the results?
|
Yes |
|
3. Point-by-point response to Comments and Suggestions for Authors This is an interesting manuscript to elucidate the associations of diabetes and hyperglycemia with length of stay of acute burn patients. The manuscript can be considered for publication after addressing the following comments.
Re: The authors kindly thank the reviewer for their appraisal and endorsement.
|
||
Comments 1: In the abstract, what is the different between admission and during admission in the statement "(admission 10.3 vs 11.5 mmol/L; during admission 9.9 vs 9.8 20 mmol/L)"?
|
||
Response 1: Thank you identifying the lack of distinction in our used terminology. We acknowledge that they do not clearly describe the difference between measured values. ‘admission BGL’ refers to pinprick blood glucose at time of admission, whilst ‘during admission’ describes the BGL measurements collected during inpatient stay. I have updated the abstract, and the manuscript contents, to clearly state BGL ‘during inpatient stay’, rather than ‘during admission’ to avoid the misinterpretation.
Changes can be found on lines 115, 117 and 168 as well as in Table 2.
|
||
Comments 2: Use either BGL or BSL, not both, consistently in the manuscript.
|
||
Response 2: Many thanks for identifying this. The authors have elected to use BGL consistently throughout the manuscript. We have changed all usage of ‘BSL’ to ‘BGL’.
Changes can be found in Tables 1, 2, 3 and 4
|
||
Comments 3: In the table 1 or in a supplementary table, the authors should indicate the number and percentage of patients in each group with BSC >11.1, >16, and <4.
|
||
Response 3: Thank you for your recommendation. The authors agree that indicating the number/percentage of patients contributing to hypoglycaemic and hyperglycaemic episodes will better characterize the quality of BGL control across the population. We have now added descriptive statistics of the groups contributing to each BGL measurements of <4, >=11.1 and >16 mmol/L.
Changes can be found in Table 1 where data is presented as a percentage of patients experiencing episodes, with total counts following in brackets.
|
||
Comments 4: In table 2, are there any patients with known diabetes but did not experienced hyperglycemia?
|
||
Response 4: The authors appreciate the question posed above. Table 2 is an extension of Table 1, looking specifically at subgroups within previously established cohorts that experience hyperglycaemia (BGL >= 11.1 mmol/L). The number of individuals with Known Diabetes experiencing hyperglycaemia in Table 2 (n=24) is derived from the population of those with Known Diabetes in Table 1 (n=34). This allocation is exhaustive with binary outcomes, indicating that those excluded from Table 2 (n=10) did not experience hyperglycaemia.
At this stage, the authors have not added additional rows in Table 2, however please advise if this presentation remains unclear. |
||
4. Response to Comments on the Quality of English Language |
||
Point 1: English language fine. No issues detected |
||
Response 1: The authors extend their thanks for the reviewer’s assessment.
|
||
5. Additional clarifications |
||
Nil |

Reviewer 2 Report
Comments and Suggestions for Authors.
Author Response
1. Summary |
|
|
Thank you very much for taking the time to review this manuscript.
Re: The authors kindly thank the reviewer for their appraisal and endorsement.
|
||
2. Questions for General Evaluation |
Reviewer’s Evaluation |
Response and Revisions |
Does the introduction provide sufficient background and include all relevant references? |
Yes |
The authors thank the reviewer for their kind assessment. |
Are all the cited references relevant to the research? |
Yes |
|
Is the research design appropriate? |
Yes |
|
Are the methods adequately described? |
Yes |
|
Are the results clearly presented? |
Yes |
|
Are the conclusions supported by the results? |
Yes |
|
3. Point-by-point response to Comments and Suggestions for Authors |
||
Nil comments. The authors thank the reviewer for their appraisal.
|
||
4. Response to Comments on the Quality of English Language |
||
Point 1: English language fine. No issues detected |
||
Response 1: The authors extend their thanks for the reviewer’s assessment.
|
||
5. Additional clarifications |
||
Nil.
|

Reviewer 3 Report
Comments and Suggestions for Authors
The authors present a very nicely constructed article regarding diabetes and hyperglycemia in burn patients. I hope that in the future they can investigate and publish the relationship between their findings and healing time.
I only have minor comments:
Abstract
Line 13: burns instead of burn?
Tabels and figures
Please provide a legend for abbreviations
Author Response
1. Summary |
|
|
Thank you very much for taking the time to review this manuscript. Please find the detailed responses below and the corresponding revisions in tracked changes in the re-submitted files, alongside a finalized manuscript incorporating all reviewer recommendations.
|
||
2. Questions for General Evaluation |
Reviewer’s Evaluation |
Response and Revisions |
Does the introduction provide sufficient background and include all relevant references? |
Yes |
The authors thank the reviewer for their kind assessment. |
Are all the cited references relevant to the research? |
Yes |
|
Is the research design appropriate? |
Yes |
|
Are the methods adequately described? |
Yes |
|
Are the results clearly presented? |
Yes |
|
Are the conclusions supported by the results?
|
Yes |
|
3. Point-by-point response to Comments and Suggestions for Authors |
||
The authors present a very nicely constructed article regarding diabetes and hyperglycemia in burn patients. I hope that in the future they can investigate and publish the relationship between their findings and healing time. I only have minor comments. Re: The authors kindly thank the reviewer for their appraisal and recommendations.
Comments 1: Abstract. Line 13: burns instead of burn?
|
||
Response 1: The authors thank the reviewer for the lexical recommendation. Unfortunately, Line 13 does not yet include the abstract. The authors have interpreted the directed Abstract Line 13 to be manuscript line 26. Here, the authors are unsure of the proper grammatical usage in ‘extent of burns’ or ‘extent of burn’. Therefore, we have altered the manuscript to read ‘extent of burn injury’.
Changes can be found in manuscript line 26. |
||
Comments 2: Tables and figures. Please provide a legend for abbreviations |
||
Response 2: The authors thank the reviewer for their recommendations. We agree that a legend for abbreviations will assist readers in interpreting presented data.
Beneath Table 1, the following abbreviations have been listed. - BGL: Blood glucose, - TBSA: Total body surface area, - NGT: Nasogastric tube, - LOS: length of stay These abbreviations are again used throughout Table 2-4, however, are not relisted. Please advise if the reviewer prefers these to be subsequently listed.
The title of Figure 2, has been updated to eliminating abbreviations, now reading ‘Receiver Operating Characteristic Curve for Above Median Length of Stay’ Beneath Figure 2, the following abbreviation has been listed - AUC: Area under the Reciver Operating Characteristic Curve
|
||
4. Response to Comments on the Quality of English Language |
||
Point 1: English language fine. No issues detected |
||
Response 1: The authors extend their thanks for the reviewer’s assessment.
|
||
5. Additional clarifications |
||
Nil.
|

Reviewer 4 Report
Comments and Suggestions for Authors
This is a well written paper which may be of interest to clinicians involved in burns treatment. However, I think the data requires analysis by a professional statistician.
Two obvious points are:
1. dividing LOS into above and below median is inappropriate. Continuous variables. including "survival" variables and other durations should - if at all possible - be analyzed as such.
2. That LOS is associated with inpatient hyperglycaemia can be explained by bias: the longer the LOS the greater the possibility that an event occurs during their stay. This would even include trivially unrelated events such as birthdays of spouses. Proper analysis should avoid such bias.
Author Response
1. Summary |
|
|
Thank you very much for taking the time to review this manuscript. Please find the detailed responses below and the corresponding revisions in tracked changes in the re-submitted files, alongside a finalized manuscript incorporating all reviewer recommendations.
|
||
2. Questions for General Evaluation |
Reviewer’s Evaluation |
Response and Revisions |
Does the introduction provide sufficient background and include all relevant references? |
Yes |
|
Are all the cited references relevant to the research? |
Yes |
|
Is the research design appropriate? |
Can be improved |
|
Are the methods adequately described? |
Yes |
|
Are the results clearly presented? |
Yes |
|
Are the conclusions supported by the results?
|
Can be improved |
|
3. Point-by-point response to Comments and Suggestions for Authors This is a well written paper which may be of interest to clinicians involved in burns treatment. However, I think the data requires analysis by a professional statistician. Re: The authors kindly thank the reviewer for their appraisal and recommendations.
|
||
Comments 1: dividing LOS into above and below median is inappropriate. Continuous variables. including "survival" variables and other durations, should - if at all possible - be analyzed as such.
|
||
Response 1: The authors thank the reviewer for the above suggestion. Our initial analyses found continuous modelling of length of stay to be suboptimal, and a potential limitation of our study. When we modelled the data using Poisson regression, we noticed large variance to mean value ratios (overdispersion), with potential inaccuracy of estimated standard errors. Our current model dichotomised the LOS outcome in order to address the skewed nature of the data. As there is no generally accepted definition of extended LOS in this context, we used the median value (5 days) for the analysis.
We have amended the Methods to clarify that LOS was dichotomised using the median value for the purpose of this analysis (lines 104-105). We have noted the analysis of LOS as a dichotomized variable as a limitation of the study in the Discussion (lines 336-337).
We hope this response satisfies your recommendations.
|
||
Comments 2: 2. That LOS is associated with inpatient hyperglycaemia can be explained by bias: the longer the LOS the greater the possibility that an event occurs during their stay. This would even include trivially unrelated events such as birthdays of spouses. Proper analysis should avoid such bias.
|
||
Response 2: The authors appreciate the identification of potential study bias. The key finding from the LOS analysis is its association with extent of burn injury. As TBSA is determined on presentation to hospital, our opinion is that the bias described above would not apply.
We have added to the Discussion to emphasize the observational nature of the analysis, and to flag the possibility that the association with inpatient hyperglycemia may be coincidental (lines 337-338). We have also amended the conclusion to highlight that the major finding from the LOS analysis is the association with extent of burn injury (lines 344-345).
We hope our details help clarify our assessment of the identified bias satisfactorily.
|
||
4. Response to Comments on the Quality of English Language |
||
Point 1: English language fine. No issues detected |
||
Response 1: The authors extend their thanks for the reviewer’s assessment.
|
||
5. Additional clarifications |
||
Nil.
|

Round 2
Reviewer 4 Report
Comments and Suggestions for Authors
I am still unhappy with the statistical analysis. Overdispersion is common in "Poisson" regression and methods have been developed to deal with it.
Author Response
1. Summary |
|
|
Thank you very much for taking the time to review this manuscript. Please find the detailed responses below and the corresponding revisions in tracked changes in the re-submitted files, alongside a finalized manuscript incorporating all reviewer recommendations.
|
||
2. Questions for General Evaluation |
Reviewer’s Evaluation |
Response and Revisions |
Does the introduction provide sufficient background and include all relevant references? |
Yes |
|
Are all the cited references relevant to the research? |
Yes |
|
Is the research design appropriate? |
Can be improved |
|
Are the methods adequately described? |
Yes |
|
Are the results clearly presented? |
Yes |
|
Are the conclusions supported by the results?
|
Must be improved |
|
3. Point-by-point response to Comments and Suggestions for Authors This is a well written paper which may be of interest to clinicians involved in burns treatment. However, I think the data requires analysis by a professional statistician. Re: The authors kindly thank the reviewer for their appraisal and recommendations.
|
||
Comments 1: dividing LOS into above and below median is inappropriate. Continuous variables. including "survival" variables and other durations, should - if at all possible - be analyzed as such. Review 2 Comment 1: I am still unhappy with the statistical analysis. Overdispersion is common in "Poisson" regression and methods have been developed to deal with it.
|
||
Response 1: The authors thank the reviewer for their recommendations of alternative statistical analysis methods. The authors have now considered hospital length of stay as count data. This was initially modelled through Poisson regression, however a dispersion parameter of 2.91 was suggestive of overdispersion with potential inaccuracy of estimated standard errors. A Quasipoisson regression was subsequently modelled but similarly returned a dispersion parameter of 2.91. A negative binomial regression model was then fitted with Chi square value of 317.1 when compared against the Poisson model, suggestive of being a more appropriate model. All models were compared using a final multivariate model, created by backward selection.
The final multivariate negative binomial regression model can be reviewed on the amended Table 4. Interpretations and conclusions were updated in the abstract (lines 26-28, and 30-31), Results (line 193-209), and Discussion (Line 277-283, and 331-335)
Figure 2 (ROC Curve) was removed as there was no longer a dichotomous model that required accuracy assessment.
|
||
4. Response to Comments on the Quality of English Language |
||
Point 1: English language fine. No issues detected |
||
Response 1: The authors extend their thanks for the reviewer’s assessment.
|
||
5. Additional clarifications |
||
Nil. |
